# Silicone Breast Implant Coated with Triamcinolone Inhibited Breast-Implant-Induced Fibrosis in a Porcine Model

**DOI:** 10.3390/ma14143917

**Published:** 2021-07-14

**Authors:** Sun-Young Nam, Han Bi Ji, Byung Ho Shin, Pham Ngoc Chien, Nilsu Donmez, Xin Rui Zhang, Beom Kang Huh, Min Ji Kim, Young Bin Choy, Chan Yeong Heo

**Affiliations:** 1Department of Plastic and Reconstructive Surgery, Seoul National University Bundang Hospital, Seongnam 13620, Korea; 99261@snubh.org (S.-Y.N.); shinzmatt@naver.com (B.H.S.); ngocchien1781@gmail.com (P.N.C.); nlsdonmez@gmail.com (N.D.); zhangxinrui@snu.ac.kr (X.R.Z.); 2Interdisciplinary Program in Bioengineering, College of Engineering, Seoul National University, Seoul 08826, Korea; hanbi2697@snu.ac.kr (H.B.J.); bkhuh85@snu.ac.kr (B.K.H.); kmj346@snu.ac.kr (M.J.K.); 3Department of Plastic and Reconstructive Surgery, College of Medicine, Seoul National University, Seoul 03080, Korea; 4Medical Research Center, Institute of Medical & Biological Engineering, Seoul National University, Seoul 03080, Korea; 5Department of Biomedical Engineering, College of Medicine, Seoul National University, Seoul 03080, Korea

**Keywords:** triamcinolone, capsular contracture, silicone implant, drug delivery, fibrosis

## Abstract

Cosmetic silicone implants for breast reconstruction often lead to medical complications, such as abnormally excessive fibrosis driven by foreign body granulomatous inflammation. The purpose of this study was to develop a silicone breast implant capable of local and controlled release of a glucocorticoid drug triamcinolone acetonide (TA) for the prevention of silicone-breast-implant-induced fibrosis in a Yorkshire pig model (in vivo). Implants were dip-coated in a TA solution to load 1.85 μg/cm^2^ of TA in the implant shell, which could release the drug in a sustained manner for over 50 days. Immunohistochemical analysis for 12 weeks showed a decline in tumor necrosis factor-α expression, capsule thickness, and collagen density by 82.2%, 55.2%, and 32.3%, respectively. Furthermore, the counts of fibroblasts, macrophages, and myofibroblasts in the TA-coated implants were drastically reduced by 57.78%, 48.8%, and 64.02%, respectively. The TA-coated implants also lowered the expression of vimentin and α-smooth muscle actin proteins, the major profibrotic fibroblast and myofibroblast markers, respectively. Our findings suggest that TA-coated silicone breast implants can be a promising strategy for safely preventing fibrosis around the implants.

## 1. Introduction

Silicone breast implants are used for esthetical and reconstructive purposes in the public health field. However, capsular contracture (CC), which induces implant deformities and pain in an advanced stage, is the most severe side effect with the use of silicone breast implants [1,2]. Symptoms range from breast hardening and discomfort to complete breast and volume collapses due to excessive production of a fibrous capsule around the implant [3]. CC has an occurrence rate of 8% to 45% [4]. Therefore, CC remains a significant challenge in plastic surgery and is one of the most frequent postoperative complications connected with revision and implant removal during alloplastic breast reconstruction or breast enlargement [5]. The complications in CC contribute to patient dissatisfaction with reconstructive breast implant surgery [6].

The main cause of capsular contracture is excessive fibrosis caused by an abnormally highly regulated and long-lasting inflammatory response. Usually after silicone breast implant insertion, acute inflammation starts with the recruitment of inflammatory cells (e.g., neutrophils and monocytes), which promotes the release of cytokines, such as tumor necrosis factor (TNF)-α, interleukin (IL)-1β, and other signaling factors from the tissues around the implant [7,8,9]. Due to the persistent involvement of the implant, the inflammatory response reaches the chronic stage, in which monocytes differentiate into macrophages responsible for the release of transforming growth factor-β (TGF-β) and other pro-inflammatory cytokines to increase the distribution and activation of fibroblasts [9,10]. These fibroblasts synthesize excessive collagen and also differentiate into myofibroblasts, which cause mechanical tension in the peripheral collagen tissue around the implant and thus induce CC [10,11].

Several measures are recommended to reduce the incidence of CC, such as surface modifications of silicone breast implants with micro-/nanotextured materials, anti-adhesive and/or antibacterial materials, and drug coating using triamcinolone, tranilast, montelukast, and zafirlukast, which have reported varying levels of success [3,12]. For drug coating, steroids are used commonly to promote anti-inflammatory activity, and glucocorticoids are considered good candidates for the prevention of CC caused by overactive inflammation [13,14]. Glucocorticoids can prominently suppress the overall inflammatory cytokines, such as TNF-α and IL-1β, due to the inhibition of transcription factors in inflammatory cells [15,16,17]. They can also suppress the expression of chemoattractants and adhesion molecules, which play a key role in the recruitment of inflammatory cells [14].

Triamcinolone (TA) is an FDA-approved medication that is normally administered not continuously but rather as a bolus through local injection. Therefore, TA can be a good optional drug for reducing fibrosis when accompanied with a suitable carrier [3,4]. The local, sustained supply of such an anti-inflammatory drug around a silicone breast implant would therefore be useful to modulate the upregulated inflammation and thus avoid capsular contracture [18]. Continuous drug exposure during acute inflammation inhibits the recruitment and activation of polymorphonuclear cells (PMNs) and monocytes. This results in fewer macrophages and TGF-β downregulation, which are otherwise highly expressed or developed during the chronic inflammation process. This aids in the inhibition and spread of activated fibroblasts, resulting in less collagen synthesis [11,19,20].

Several studies have reported using animal models, such as mice, rabbits, and pigs, with different outcomes, but the consensus is that rats provide an effective histological extrapolation of human tissue as a reproducible and low-cost effective model [12]. However, in the current study, we investigated the reduction in CC in a porcine model with a TA-coated silicone breast implant. The capsule thickness, collagen density, inflammation score, muscle thickness, and the count of fibroblasts, myofibroblasts, and macrophages were estimated with our proposed Yorkshire pig model to assess the effect of TA-coated silicone breast implants on CC reduction. Further, TGF-β and α-SMA production was also estimated by Western blot.

## 2. Materials and Methods

### 2.1. Materials

The silicone breast implants (SFS-LP) were a generous gift from Hans Biomed (Seoul, Korea). TA was purchased from Tokyo Chemical Industry (Japan), isopropyl alcohol (IPA) and dimethylformamide (DMF) from DaeJung (Siheung-si, Gyeonggido, Korea), acetonitrile (ACN) from J. T. Baker (Billerica, MA, USA), and Tween 80 from Merck (Kenilworth, NJ, USA). Paraformaldehyde (4%) and isoflurane were supplied by Dreamcell (Seoul, Korea) and Hana Pharm (Seoul, Korea), respectively. Xylene, ethanol, and acetic acid solution (1%) were purchased from Duksan Pure Chemicals (Seoul, Korea). Phosphate-buffered saline (DPBS) was supplied by the Seoul National University Biomedical Research Institute (Seoul, Korea) and Hoechst 33342 for nucleus staining by Invitrogen (Carlsbad, CA, USA). For cell culture, DMEM/high glucose, FBS, and penicillin/streptomycin were purchased from Hyclone (Logan, UT, USA).

For in vivo evaluation, Zoletil 50 was purchased from Virbac (Fort Worth, TX, USA) and Rompun from Bayer (Leverkusen, Germany). Paraformaldehyde (4%) was purchased from KCFC (Seoul, Korea). For H&E staining, xylene, ethanol, and hydrochloric acid (35–37%) were purchased from Duksan Pure Chemicals (Ansan, Korea). Ammonia solution (28–30%) was obtained from Junsie Chemical (Tokyo, Japan). Modified Mayer’s H&E Y solutions were supplied by Richard-Allan Scientific (Kalamazoo, MI, USA). For MT staining, acetic acid (1%) was obtained from Duksan Pure Chemicals (Ansan, Korea). Biebrich scarlet-acid fuchsin, phosphomolybdic acid, and aniline blue solutions were purchased from Sigma-Aldrich (St. Louis, MO, USA); immunofluorescence staining solution (10×) from Dako (Glostrup, Denmark); anti-TNF-α (ab1793) and Alexa Fluor 488 (au11059) from Abcam and Invitrogen, respectively;anti-TGF-β (sc-146) from Santa Cruz Biotechnology (Dallas, TX, USA); and anti-vimentin (ab92547), anti-α-SMA (ab5694), β-actin, and CD68 from Abcam (Cambridge, MA, USA). Paraffin was supplied by Merck (Kenilworth, NJ, USA).

### 2.2. Preparation of Silicone Breastimplant Samples

Two different implant samples were prepared with silicone breast implants in clinical use (Bellagel, Hans Biomed, Seoul, Korea): intact implants without the drug (control group) and implants coated with TA (TA group). To prepare the TA coating, a solution of TA (0.05% *w*/*v*) was prepared in IPA. Then, the implants were fully immersed in 250 mL of the resulting drug solution and coated for 10 min at room temperature with shaking at 125 rpm. The coated implants were then washed for 10 s and dried at 70 °C for 2 h.

### 2.3. In Vitro Drug Release Experiments

The TA-coated implants were tested for drug release under in vitro conditions. Briefly, each implant was immersed in 200 mL of phosphate-buffered saline (1X PBS, pH 7.4) containing 0.1% *w*/*v* Tween 80 to meet the sink condition of TA and incubated at 37 °C and 125 rpm in a shaking incubator (SI-600R; Jeio Tech, Seoul, Korea). At predetermined periods, the release medium was fully extracted and an equal volume of fresh medium was added back to the solution. The obtained media were each tested by HPLC-MS, as previously described [21]. The experiment was performed in triplicate for statistics.

### 2.4. Yorkshire Pigs for in Vivo

Yorkshire white pigs, with an average weight of approximately 15 kg (Optipharm, Cheongju, Korea), aged 8 weeks, were assigned for in vivo testing. Each pig was lodged separately in a cage for about 2–4 weeks to be accustomed to adequate food, illumination, temperature, and humidity under standard conditions until in vivo experiments. Treatment for animals was performed according to the protocol approved by the Seoul National University Bundang Hospital Institutional Animal Care and Use Committee (approval number: BA 1830-243/024-01).

### 2.5. Implantation Procedures

For in vivo evaluation, two different types of silicone breast implants, control (only silicone implants) and TA-coated implants, were used. Twenty-four implants were used for each group. After grafting, each implant was randomized and double-blinded for further analysis. A total of 48 implants were inserted into 12 pigs (4 implants per pig). For implantation, the pigs were anesthetized with an intramuscular injection of 10 mg/kg of ketamine (Yuhan, Seoul, Korea) with 2 mg/kg of xylazine (Bayer, Ansan, Korea) under general anesthesia. In addition, the surgical sites were anesthetized with an intramuscular injection of 10% povidone-iodine solution (Firson, Cheonan-si, Korea) and cefazoline (60 mg/kg) (Chong Kun Dang, Seoul, Korea). Next, four separate vertical incisions were made 5 cm inside both nipple lines. The implants were inserted into the thoracic region under the skin and thin panniculus carnosus muscle. After insertion, the skin incisions were stitched up using 4-0 polyglactin suture (Vicryl, Ethicon, Somerville, NJ, USA). The pigs were then transferred back to the animal facility to monitor their well-being and any possible infection every day.

### 2.6. Implant Explantation and Tissue Processing

Six months after implantation, the pigs were sacrificed under the guidelines of the American Veterinary Medical Association (AVMA) by an injection of potassium chloride (KCl) (JW Pharmaceutical Corporation, Seoul, Korea). A skin incision was made to extract the silicone breast implants, including the surrounding capsule tissues. A total of 48 tissue samples from 48 implants were; therefore, collected. The capsule tissues were immediately fixed in 10% formalin until further analysis. The fixed tissue samples were washed for 12 h with deionized water and dehydrated for 4 h in xylene (Samechun, Seoul, Korea) from 70% to 99.9%. The samples were then embedded in paraffin using a tissue embedding center (Leica, Wetzlar, Germany) and cut through 5 μm using a Leica RM2255 rotary microtome (Leica, Wetzlar, Germany).

### 2.7. In Vivo Evaluation of Capsule Thickness and Collagen Density

The paraffin tissue cubes were cut into 4 mm-thick slices. To remove the diaphragm, chemicals such as xylene and ethanol were used. The capsule thickness, collagen density, and number of fibroblasts and myofibroblasts were estimated, as previously described [20]. Hematoxylin and eosin (H&E) were used to stain the tissues, and the capsule thickness was determined at 40× magnification under a microscope (LSM 700; Carl Zeiss, Oberkochen, Germany). The capsules were described from top to bottom of the dorsal subcutaneous muscle. The total capsule thickness was randomly photographed in three different parts, and ImageJ (version 1.47 software, National Institutes of Health, USA) [20] was used to calculate the capsule thickness.

Masson’s trichrome (MT) staining was used to determine the amount of collagen deposition throughout the implants [21]. The collagen was dyed blue; thus, the blue region of each image was selectively determined using ImageJ software (version 1.47 software, National Institutes of Health, USA) [20]. The selected region was divided by the entire area of tissue in the same image to obtain the collagen density percentile value.

### 2.8. Estimation of Fibrosis and Inflammation Scores in the Sliced Tissues

To count the cells involved in fibrosis (i.e., fibroblasts, macrophages, and myofibroblasts), immunofluorescence (IF) staining was performed. Primary anti-TNF-α, anti-CD68, anti-vimentin, and anti-α-SMA mouse antibodies were used [22]. Anti-CD68 rabbit and anti-TNF-α antibodies diluted to 1:300 and 1:100, respectively, were used for macrophages. Anti-α-SMA mouse and anti-vimentin antibodies diluted to 1:50 and 1:250 times, respectively, were used for myofibroblasts and fibroblasts, respectively. In addition, secondary anti-rabbit and anti-mouse antibodies dilution from 1:2000 were used to determine fluorescence. Only the anti-mouse secondary antibody for anti-TNF-α was diluted to 1:1000. All dilution was performed using sterile 1X PBS with 1% bovine serum albumin (*w*/*v*) and 0.1% SDS (*v*/*v*). The slides were mounted and stained in VECTASHIELD mounting medium in DAPI (H-1200; Vector Laboratories, Burlingame, CA, USA). Fibrosis-related cells were counted with an image area of 0.48 mm^2^ (200× magnification). Three points on each image were randomly chosen in the capsule region. The inflammation scores were assessed, as previously described [20], and graded from 0 to 3 (none, mild, moderate, and severe, respectively).

### 2.9. Western Blot Analysis of Specific Markers Related to CC

For insight into the factors closely associated with capsule formation following the insertion of silicone breast implants into the tissue, Western protein expression blot analysis was performed for specific fibrosis markers, such as α-SMA and TGF-β1. The protocol for Western blot was followed, as previously described [22]. α-SMA and TGF-β1 protein expression levels in capsular tissue were determined using ImageJ software (version 1.47 software, National Institutes of Health, USA) [20], quantified with the total gray values for each band and then normalized to the respective β-actin. All experiments were performed in triplicate, and the mean values were noted.

### 2.10. Statistical Analysis

All statistical analyses were performed using GraphPad Prism version 5.0 (GraphPad Software, San Diego, CA, USA) [23]. The data were represented as the mean ± standard error of the mean (SEM). The unpaired t-test, one-way ANOVA, and Tukey’s multiple-comparison tests between TA-coated and uncoated silicone groups were performed to determine significant differences in the in vivo experiments. The degree of significant difference was indicated as the *p*-value. Asterisks on the graphs shown as ****, ***, **, and * indicated *p* < 0.0001, 0.0001 ≤ *p* < 0.001, 0.001 ≤ *p* < 0.01, and 0.01 ≤ *p* < 0.05, respectively.

## 3. Results

### 3.1. Characterization of TA Coating

The TA-coated silicone breast implants were developed in a porcine model to treat CC. In our study, we coated the implants by soaking them in a TA solution (Figure 1), with a loading amount of 1.85 ± 0.1 μg/cm^2^ (Table 1). We performed an in vitro drug release study with the TA-coated implants, as shown in Figure 2. There was an apparent initial burst release for the first three days (ca. 0.67 μg/cm^2^) due to the TA adsorbed near the implant surface. Subsequently, TA was continuously released at a rate of 0.015 μg/cm^2^ per day, and approximately 70% of TA was released until the end of 50 days.

### 3.2. Capsule Thickness and Collagen Density

To investigate the antifibrotic efficacy of triamcinolone, we assessed the capsule thickness and collagen density around the implant samples. As shown in Figure 3a, the capsule thickness was determined between the two animal groups (i.e., control (only silicone implants) and TA (TA-coated silicone implants) groups). The capsule thickness was lower in the TA group compared with the control group, and the difference was significant over the entire test period (*p* < 0.001, Figure 3b). The capsule thickness was 5199.2 µm after 12 weeks and then decreased to 2328.9 µm in the TA group (Figure 3b). However, the capsule thickness in the control group increased by 61.8% at 12 weeks compared to four weeks. A reduction of 55.2% was noticed in the capsule thickness at 12 weeks in the TA group compared with the control group (Figure 3b).

As displayed in Figure 3c, MT staining clearly showed a reduction in the collagen density in the TA-coated silicone biomaterial, which reduced fibrosis. As shown in Figure 3d, the collagen density reduced by 32.4% in the TA group, which was statistically significant over the entire testing period. The collagen density in the TA group reduced from 78.3% to 56.9% at 12 weeks, while that in the control group reduced slightly by 7.06% (*p* < 0.001, Figure 3d). After 12 weeks, the collagen density in the control and TA groups was 83.6% ± 1.6% and 58.7% ± 5.3%, respectively (*p* < 0.05, Figure 3d). These results suggest that TA-coated silicone breast implant could have an impact on capsular contracture by reducing fibrosis.

### 3.3. Inflammation Score and Macrophages

The overall inflammation reduced in the TA group compared with the control group (Figure 4a), suggesting prolonged anti-inflammatory activity of TA in the implanted area. H&E staining images showed reduced inflammation in the TA group after 12 weeks (Figure 4a). The inflammation score reached its maximum after eight weeks in the control group and gradually reduced by 48.2% after 12 weeks (*p* < 0.001). However, a reduction of 37.3% from weeks four to 12 was noted in the TA group (Figure 4b). In the control group, the inflammation was moderate, while in the TA group, it was mild.

The mean values of macrophage cell counts were estimated in CD68-positive cells and compared to the control group. Figure 4c,d shows a significantly lower number of macrophages in the TA group (*p* < 0.001). The macrophage count increased at week eight in the control group compared to week four; However, a gradual reduction of 9.25% was noted by week 12 compared to week eight (Figure 4d). In the TA group, the macrophage count declined by 10.04% from week four to week 12 (Figure 4d). The CD68-positive cells clearly showed a reduction in the macrophage cell count in the TA group compared with the control group at four, eight, and 12 weeks (Figure 4d).

### 3.4. Fibroblasts and Myofibroblasts

TGF-β is well known to contribute to the development of silicone-implant-induced fibrosis. TGF-βpromotes collagen synthesis and α-SMA expression in fibroblasts. In the current study, TGFβ1 and α-SMA were expressed in lower concentrations in TA group (Appendix A). When the relative expression levels of TGF-β1 and α-SMA were calculated normalized to β-actin, α-SMA (Appendix A) and TGF-β1 (Appendix A) showed a decline in expression, suggesting the anti-inflammatory property of TA.

Similarly, a drastic decline in the fibroblast cell count by 58.67% was noted in the TA group by the end of 12 weeks compared to four weeks (Figure 5a). The fibroblast count decreased by 15.6% in the control group from weeks eight to 12. Vimentin-stained cells suggested a gradual decline in fibroblasts in the TA group (Figure 5b). In addition, by the end of 12 weeks of implantation, a significant reduction of 57.78% in fibroblasts was noted in the TA group compared with the control group.

The myofibroblast count gradually increased by 62.9% in the control group from weeks four to 12, while it declined by 16.6% in the TA group by the end of 12 weeks (Figure 5c). α-SMA-positive cells suggested a clear decline in myofibroblasts in the TA group compared with the control group (Figure 5d).

## 4. Discussion

The present work focused on the design of TA-coated silicone breast implants that were incorporated into a Yorkshire pig model to reduce the induced capsular contracture. The implants were designed to be successfully inserted into the subcutaneous space and to release TA in a sustained manner into the incorporated membrane that supports fibrosis. The implants were characterized in terms of their drug release, capsule thickness, collagen density, counts of cells involved in fibrosis, and expression of protein markers associated with fibrosis in the proposed model.

In this work, we used a TA solution prepared in an organic solvent, IPA, to dip-coat the shell of the implant. The drug solution was thus well absorbed into the silicone matrices that could act as the mediator of drug diffusion [20]. Jeon et al. [24] documented the use of a silicone breast implant capable of reducing fibrosis by local, continuous release of TA, a glucocorticoid. They indicated that the prepared TA-coated silicone breast implant would release TA over 12 weeks and would be able to suppress pro-inflammatory factors compared with the uncoated, intact implant. However, in our current study, even after seven weeks (50 days) of tests, the TA release was successful, with over 0.8% release per day. Thus, the anti-fibrotic effect of TA, when implanted in living animals, is evident over the entire implant surface for both large and small implant specimens [20].

H&E staining showed a clear reduction in the capsule thickness in the TA group from four to 12 weeks, while the capsule thickness increased in the control group. In the breast capsular tissue, various cells are found to be predominant. The number of fibroblasts and the collagen layer thickness correspond to Baker contraction grades [25]. The collagen density in the capsule is recognized as one of the major factors in deciding the capsule stiffness [26]. There was a prominent decline in the capsule thickness and collagen density in the TA group, which was statistically significant. This finding indicates that TA is released to the entire surrounding region of the implant with equal efficacy. Other biomaterials, including prednisolone, acellular porcine dermis (APD), a silicone drug-delivery net, also possess anti-fibrotic activity, with a decline in capsule thickness and collagen density [20,27,28]. Expression levels of the pro-inflammatory cytokine TNF-α were determined by immunofluorescence staining to understand the role of TA in the degree of inflammation in the implantation area. Furthermore, the tissue expression of myofibroblasts (α-SMA), fibroblasts (vimentin), and markers of macrophage (CD68), which are closely correlated with the CC formation, was investigated via immunohistochemical staining [29]. A significantly larger initial burst of TA release was seen in the TA group on the first day (as shown in Figure 1), but the solubilized TA released in the early stage was expected to be rapidly cleared from the implant site. The dosage and duration of delivery by the silicone breast implant should be considered successful to accurately decrease fibrosis. Abnormal silicone-implant-related fibrosis is primarily present at the late inflammation point, which has a significant impact on the level of sustained acute inflammation at the early stage [20,22]. For this reason, we examined whether TA-coated silicone breast implants downregulate the expression of the inflammatory cytokine TNF-α at four, eight, and 12 weeks. The results confirmed the effect of TA in silicone breast implants, showing a statically distinguishable reduction in TNF-α in all tested periods compared with the control group (Figure 3). This slight decrease in the expression level of TNF- α, correlating with fibrosis formation, supports the expectation that silicone breast implants coated with TA should form less capsular contracture. The tested pro-inflammatory parameters in rat models using silicone breast implants coated with APD showed the same effects, with significantly less CD68-positive cells in the myofibroblast layer [28]. Here, the cell counts also showed a distinct decrease in the experimental group after 12 weeks compared with the control group. Therefore, implants coated with TA decrease the degree of acute inflammation (Figure 4), which, in turn, reduces the formation of fibrotic tissue around the implants at week four.

It was noted that the TA-coated silicone breast implants reduced the number of fibroblasts, myofibroblasts, and macrophages (*p* < 0.001) in the entire surrounding tissues, suggesting a continuous, homogeneous exposure of TA from the entire surface of the implants. TA may be useful for modulating inflammation and reducing the CC thickness. Thus, as previously reported, other anti-inflammatory and anti-fibrotic compounds, such as pirfenidone, leukotriene receptor antagonists, and zafirlukast, can be good candidate drugs for CC management [12,30,31].

Differentiation of fibroblasts to myofibroblast plays a significant role in the development of fibrotic capsules around implants. This process is activated by the TGF-β1 signaling pathway, and thus, TGF-β1 is one of the key molecules in CC formation [32]. We aimed to focus on the interaction of these cell types with the silicone material, thus the gene expression of TGF-β1 and α-SMA was analyzed as a leading biomarker of fibrosis [33], which suggested that TA is a proficient drug molecule that can reduce CC formation in the studied animal model.

## 5. Conclusions

In conclusion, this study was conducted on silicone breast implants coated with a glucocorticoid, TA, for the inhibition of fibrosis in a porcine model. TA from the implant was released in a sustained manner for about 50 days. Notably, the in vivo Yorkshire pork model showed that TA-coated implants downregulate pro-inflammatory cytokines and reduce the number of macrophages, fibroblasts, and myofibroblasts. As a result, TA-coated implants significantly reduce the capsule thickness and collagen density compared with non-coated implants. In addition, these test results from the porcine model provide a suitable therapeutic window for TA useful for silicone breast implant development with regard to both safety and effectiveness. Therefore, the newly developed TA-coated silicone breast implants are a promising strategy to avoid capsular contracture.

## Figures and Tables

**Figure 1 materials-14-03917-f001:**
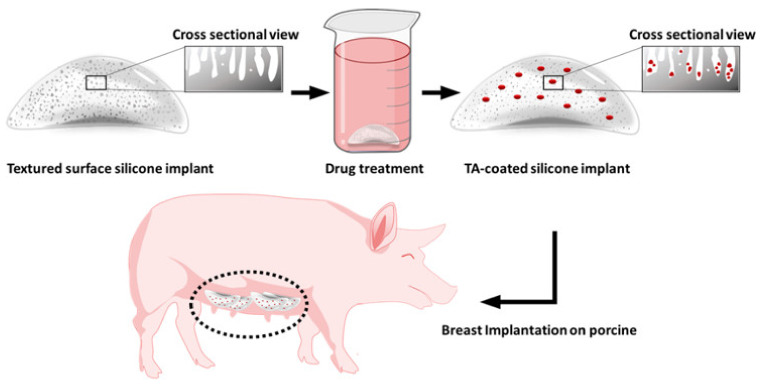
Schematic diagram for manufacturing of TA-coated silicone breast implant.

**Figure 2 materials-14-03917-f002:**
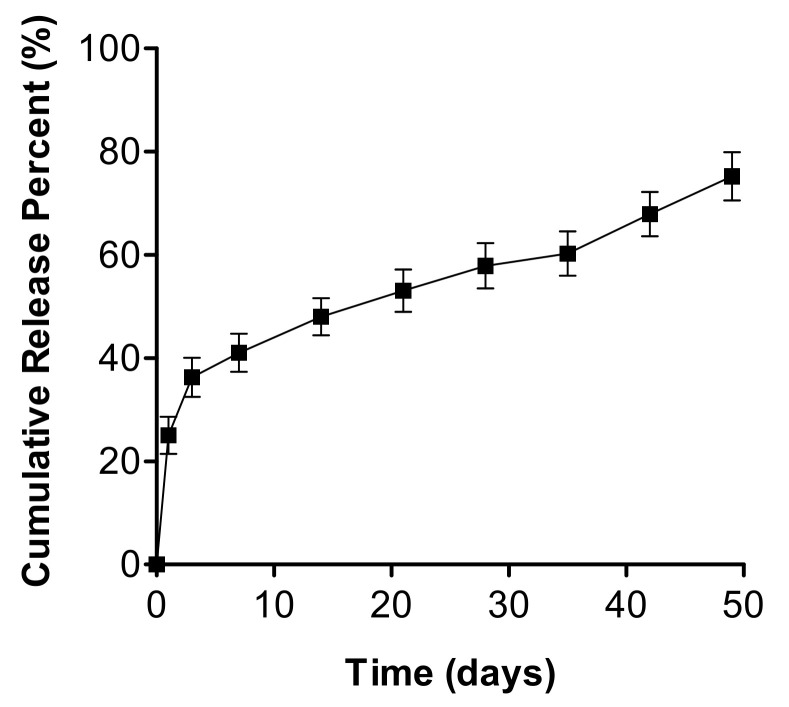
In vitro release of TA from the drug-coated silicone breast implant.

**Figure 3 materials-14-03917-f003:**
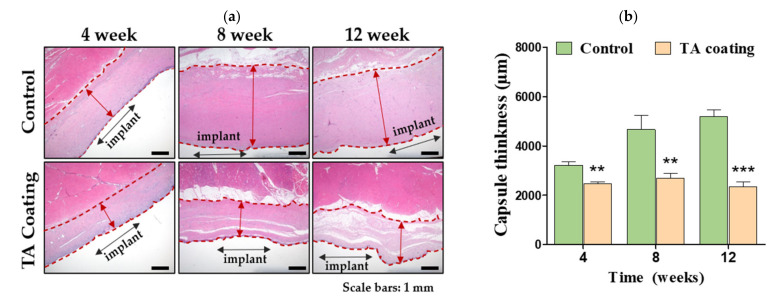
Histological evaluation of capsules formed around TA-coated and uncoated (i.e., control group) silicone breast implants. (**a**) H&E-stained images of tissues at four, eight, and 12 weeks; the red double arrow indicates the capsule thickness. Scale bars = 1 mm. (**b**) Profiles of thickness in TA-coated and uncoated silicone breast implants. (**c**) MT-stained tissue images around TA-coated and uncoated silicone breast implants at four, eight, and 12 weeks after implantation. Scale bars = 100 µm. (**d**) Profiles showing the collagen density at four, eight, and 12 weeks after implantation. ** *p* < 0.01 and *** *p* < 0.001 compared with the control group.

**Figure 4 materials-14-03917-f004:**
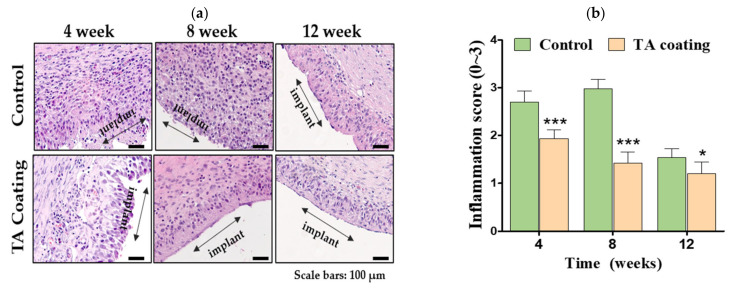
Inflammatory response to TA-coated and uncoated (i.e., control group) silicone breast implants. (**a**) Representative H&E images of the tissues at four, eight, and 12 weeks after implantation; the black double arrow shows the location of the implanted sample. Scale bars = 100 µm. (**b**) Profiles of the degree of inflammation in the capsule tissue in TA-coated and uncoated silicone breast implants (**c**) Representative image of CD68 (green) in capsule tissues obtained from the control and TA groups. Nuclei were co-stained with DAPI (blue). CD68-positive macrophages were detected using a fluorescence microscope. Scale bars = 100 µm (**d**) (d) Quantification of FITC-positive CD68-expressing cells from the control and TA groups. Data are represented as the mean ± SEM. * *p* < 0.05, ** *p* < 0.01, and *** *p* < 0.001 compared with the control group.

**Figure 5 materials-14-03917-f005:**
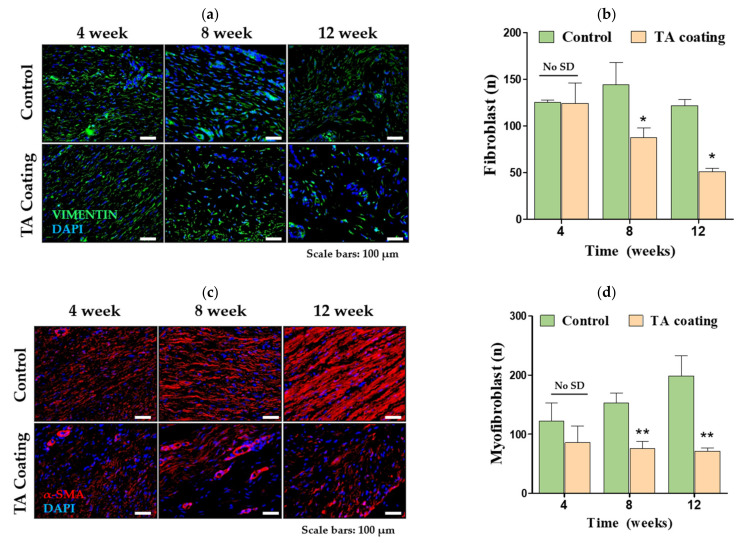
Fibrotic response to TA-coated and uncoated (i.e., control group) silicone implants. (**a**) Representative image of vimentin (green) in capsule tissues obtained from the control and TA groups. Nuclei were co-stained with DAPI (blue). Vimentin-positive fibroblasts were detected using a fluorescence microscope. Scale bars = 100 µm (**b**) Quantification of FITC-positive vimentin-expressing cells from the control and TA groups. (**c**) Representative image of α-SMA (red) in capsule tissues obtained from the control and TA groups. Nuclei were co-stained with DAPI (blue). α-SMA-positive myofibroblasts were detected using a fluorescence microscope. Scale bars = 100 µm (**d**) Quantification of TRITC-positive α-SMA-expressing cells from the control and TA groups. Data are represented as the mean ± SEM. * *p* < 0.05, ** *p* < 0.01 and compared with the control group; No SD, no significant difference.

**Table 1 materials-14-03917-t001:** Loading amount of TA after coating on the silicone breast implant.

Coating Time	TA Concentration in Coating Solution of IPA	Drug Loading Amount (μg/cm^2^)
10 min	0.05% *w*/*v*	1.85 ± 0.1

## Data Availability

Data is contained within the article or supplementary material.

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
