# Peer review of "Silicone Breast Implant Coated with Triamcinolone Inhibited Breast-Implant-Induced Fibrosis in a Porcine Model"

_materials, 2021, doi:10.3390/ma14143917_

Round 1
Reviewer 1 Report
The authors were aimed to develop a silicone breast implant capable of local and controlled release of a glucocorticoid drug, triamcinolone acetonide (TA), for the prevention of silicone breast implant induced fibrosis in a Yorkshire pig model (in vivo).
The study covers some issues that have been overlooked in other similar topics. The study was conducted with a good scientifically sound. The methodology is well described with enough experimental data and results to support the work.
Issues that need improvement: Please check English grammar and typos thorough the text.
Conclusion Section: This paragraph required a general revision to eliminate redundant sentences and to add some "take-home message".
Author Response
The authors were aimed to develop a silicone breast implant capable of local and controlled release of a glucocorticoid drug, triamcinolone acetonide (TA), for the prevention of silicone breast implant induced fibrosis in a Yorkshire pig model (in vivo).
The study covers some issues that have been overlooked in other similar topics. The study was conducted with a good scientifically sound. The methodology is well described with enough experimental data and results to support the work.
Issues that need improvement: Please check English grammar and typos thorough the text.
Answer: Thank you for your comments. Based on your comments, we have checked the grammar through the text.
Conclusion Section: This paragraph required a general revision to eliminate redundant sentences and to add some "take-home message".
Answer: Thank you for your comments. We have modified the text in the conclusion accordingly.
Reviewer 2 Report
This manuscript reports the study on the development of a silicone breast implant that is capable of local and controlled release of the drug triamcinolone acetonide, a glucocorticoid drug. The aim is to prevent silicone breast implant induced fibrosis, and the in vivo experiment was carried out in a Yorkshire pig model. The system could release the drug in a sustained manner for over 50 days. The drug coated-implants had lower expression of Vimentin and α-smooth muscle actin proteins, the major profibrotic fibroblast and myofibroblast marker, respectively. The present work claimed that this method could be a promising strategy for safely preventing fibrosis around silicone implants. This work is well written and contains useful for readers in the field. This manuscript is therefore recommended for publication after minor revision. In order to improve this manuscript, please consider the comments and suggestions, which are listed below.
- Previous related works should be mentioned, please see The impact of triamcinolone acetonide in early breast capsule formation in a rabbit model, Aesthetic Plast Surg. 2012 Aug;36(4):986-994. Silicone implants capable of the local, controlled delivery of triamcinolone for the prevention of fibrosis with minimized drug side effects, Journal of Industrial and Engineering Chemistry 2018, Volume 63, Pages 168-180; Current Approaches Including Novel Nano/Microtechniques to Reduce Silicone Implant-Induced Contracture with Adverse Immune Responses, Int. J. Mol. Sci. 2018, 19(4), 1171; https://doi.org/10.3390/ijms19041171; Silicone breast implant modification review: overcoming capsular contracture. Biomater Res 22, 37 (2018). This information is useful for readers.
- “Briefly, each sample was immersed in 200 ml of phosphate-buffered saline (PBS, pH 4)…”; what is the concentration of PBS? At 0.1M?
- “….and 3-0 ethylon (Ethicon, Inc., USA). skin cuts were stitched after injection.”; typographical error, please correct.
- “At twelfth week, the collagen density of the TA coated silicone implants was 78.3 to 56.9 %,…..”; please consider “At twelfth week, the collagen density of the TA coated silicone implants was reduced from 78.3 to 56.9 %,…..”.
Author Response
This manuscript reports the study on the development of a silicone breast implant that is capable of local and controlled release of the drug triamcinolone acetonide, a glucocorticoid drug. The aim is to prevent silicone breast implant induced fibrosis, and the in vivo experiment was carried out in a Yorkshire pig model. The system could release the drug in a sustained manner for over 50 days. The drug coated-implants had lower expression of Vimentin and α-smooth muscle actin proteins, the major profibrotic fibroblast and myofibroblast marker, respectively. The present work claimed that this method could be a promising strategy for safely preventing fibrosis around silicone implants. This work is well written and contains useful for readers in the field. This manuscript is therefore recommended for publication after minor revision. In order to improve this manuscript, please consider the comments and suggestions, which are listed below.
Previous related works should be mentioned, please see The impact of triamcinolone acetonide in early breast capsule formation in a rabbit model, Aesthetic Plast Surg. 2012 Aug;36(4):986-994. Silicone implants capable of the local, controlled delivery of triamcinolone for the prevention of fibrosis with minimized drug side effects, Journal of Industrial and Engineering Chemistry 2018, Volume 63, Pages 168-180; Current Approaches Including Novel Nano/Microtechniques to Reduce Silicone Implant-Induced Contracture with Adverse Immune Responses, Int. J. Mol. Sci. 2018, 19(4), 1171; https://doi.org/10.3390/ijms19041171; Silicone breast implant modification review: overcoming capsular contracture. Biomater Res 22, 37 (2018). This information is useful for readers.
Answer: Thank you for your suggestion and guidelines. Based on your comments, we have mentioned the previous work and added the relevant references.
“Briefly, each sample was immersed in 200 ml of phosphate-buffered saline (PBS, pH 4)…”; what is the concentration of PBS? At 0.1M?
Answer: Thank you for your comments. We used 1X PBS (Seoul National University Biomedical Research Institute, South Korea). We have added the text in the manuscript.
“….and 3-0 ethylon (Ethicon, Inc., USA). skin cuts were stitched after injection.”; typographical error, please correct.
Answer: Thank you for your comments. We have corrected the text.
“At twelfth week, the collagen density of the TA coated silicone implants was 78.3 to 56.9 %,…..”; please consider “At twelfth week, the collagen density of the TA coated silicone implants was reduced from 78.3 to 56.9 %,…..”.
Answer: Thank you for your comments. Based on your suggestion, we have changed the sentence.
Reviewer 3 Report
The paper entitled “Silicone Breast Implant Coated with Triamcinolone Inhibited 2 the Breast Implant Induced Fibrosis in a Porcine Model” covers an interesting study that deserves to be published in Materials. The study showed that silicone implants coated with the drug: triamcinolone acetonide can be used for preventing fibrosis around silicone implants. In a few places, it is unclear if the silicone implants are loaded, wrapped, or coated by the drug, therefore the description should be improved. It is unclear how the drug release is controlled and what is the release mechanisms after the implantation of the implants?
Author Response
The paper entitled “Silicone Breast Implant Coated with Triamcinolone Inhibited 2 the Breast Implant Induced Fibrosis in a Porcine Model” covers an interesting study that deserves to be published in Materials. The study showed that silicone implants coated with the drug: triamcinolone acetonide can be used for preventing fibrosis around silicone implants. In a few places, it is unclear if the silicone implants are loaded, wrapped, or coated by the drug, therefore the description should be improved. It is unclear how the drug release is controlled and what is the release mechanisms after the implantation of the implants?
Answer: We apologize for confusion. In this study, we dip-coated the silicone implant using a TA solution. Thus, the implant was immersed in the TA solution prepared in the IPA, where the IPA made the silicone shell swell and absorb the drug into the silicone matrix. As the silicone matrix would serve as the diffusion mediator, TA could be released in a sustained manner, and this type of diffusion has been already reported in many previous studies.
We have modified and added the text in the manuscript accordingly.